# The Role of Diastolic Stress Echo and Myocardial Work in Early Detection of Cardiac Dysfunction in Women with Breast Cancer Undergoing Chemotherapy

**DOI:** 10.3390/biomedicines13102341

**Published:** 2025-09-25

**Authors:** Stefanos Sokratous, Michaelia Kyriakou, Elina Khattab, Alexia Alexandraki, Elisavet L. Fotiou, Nektaria Chrysanthou, Paraskevi Papakyriakopoulou, Ioannis Korakianitis, Anastasia Constantinidou, Nikolaos P. E. Kadoglou

**Affiliations:** 1Medical School, University of Cyprus, 215/6 Old Road Lefkosias Limasol, Nicosia 2029, Cyprus; stefanossokratous94@gmail.com (S.S.); kyriakou.michailia@ucy.ac.cy (M.K.); khattab.elina@ucy.ac.cy (E.K.); korakianitis.ioannis@ucy.ac.cy (I.K.); constantinidou.anastasia@ucy.ac.cy (A.C.); 2A.G. Leventis Clinical Trials Unit, Bank of Cyprus Oncology Centre, 32 Acropoleos Avenue, Nicosia 2006, Cyprus; alexia.alexandraki@bococ.org.cy (A.A.); ppapakyr@pharm.uao.gr (P.P.); 3Bank of Cyprus Oncology Centre, Nicosia 2006, Cyprus; elisavet.l.fotiou@bococ.org.cy (E.L.F.); nektaria.chrysanthou@bococ.org.cy (N.C.); 4Laboratory of Biopharmaceutics and Pharmacokinetics, Section of Pharmaceutical Technology, Department of Pharmacy, School of Health Sciences, National and Kapodistrian University of Athens, 15771 Athens, Greece

**Keywords:** chemotherapy-related cardiac dysfunction (CTRCD), diastolic stress echocardiography (DSTE), speckle tracking echocardiography (STE), myocardial work indices, anthracycline cardiotoxicity, heart failure with preserved ejection fraction (HFpEF), cardiac biomarkers, breast cancer, global longitudinal strain (GLS)

## Abstract

**Background**: Anthracycline-based chemotherapy, while highly effective for breast cancer, poses a significant risk for chemotherapy-related cardiac dysfunction (CTRCD), mainly determined by left ventricular ejection fraction (LVEF) reduction. Objectives: We aimed to evaluate the diagnostic utility of speckle tracking analysis (STA) and Diastolic Stress Test Echocardiography (DSTE) for the early detection of cardiac dysfunction either CTRCD or heart failure with preserved ejection fraction (HFpEF) in women undergoing chemotherapy for breast cancer and developed exertional dyspnea and/or fatigue during follow-up. **Methods**: In this prospective case–control study, 133 women receiving anthracycline-based chemotherapy (with or without anti-HER2 therapy) (chemotherapy group-CTG) and 65 age-matched healthy women as the control group (CG) underwent resting echocardiographic assessment, including LVEF, global longitudinal strain (GLS), myocardial work indices, biomarkers assay (NT-proBNP, troponin, galectin-3) and DSTE at baseline. That assessment was repeated after 12 months in CTG. **Results**: In this prospective case—control study, 133 women receiving anthracycline-based chemotherapy (with or without anti-HER2 therapy) were included. Based on the presence of CTRCD, they were further subdivided into a CTRCD subgroup (*n* = 37) and a CTRCD-free subgroup (*n* = 88). At the end of this study, CTG showed worse values of LVEF, GLS, myocardial work indices than baseline and CG (*p* < 0.05). Subgroup comparison (CTRCD vs. CTRCD-free) showed significant impairment in LVEF (53.60% vs. 62.60%, *p* < 0.001), GLS (–16.68% vs. −20.31%, *p* < 0.001), DSTE-derived tricuspid regurgitation maximum velocity (TRVmax) (3.05 vs. 2.31 m/s, *p* < 0.001) and elevated biomarkers (NT-proBNP: 200.06 vs. 61.49 pg/mL; troponin: 12.42 vs. 3.95 ng/L, *p* < 0.001) in the former subgroup. Regression analysis identified GLS, NT-proBNP, troponin, and TRVmax as independent predictors of CTRCD. Notably, a subgroup of CTRCD-free patients (*n* = 16) showed a high probability for HFpEF based on the HFA-PEFF score, with elevated GLS, NT-proBNP and DSTE-derived TRVmax compared to the rest of CTRCD-free patients and the CG (*p* < 0.001). **Conclusions:** STA and DSTE significantly outperform conventional LVEF in detecting subclinical cardiac dysfunction among women with breast cancer receiving chemotherapy. The combination of novel echocardiographic techniques and biomarkers may enable the detection of early CTRCD, including the under-estimated presence of HFpEF among breast cancer women with HF symptoms.

## 1. Introduction

Breast cancer (BC) is one of the most common malignancies affecting women worldwide [1]. Anthracyclines induce cardiotoxicity primarily via inhibition of topoisomerase IIβ in cardiomyocytes, causing DNA double-strand breaks, mitochondrial injury, and excessive reactive oxygen species (ROS) production, which trigger myocyte apoptosis and fibrosis [2,3]. In contrast, HER2-targeted agents such as trastuzumab impair the neuregulin-1/HER2 pathway, which is essential for cardiac stress adaptation and contractile reserve [4]. Despite their significant anti-neoplastic efficacy anthracyclines, are among the most cardiotoxic chemotherapeutic agents [5,6]. Even novel targeted therapies, such as the monoclonal antibody trastuzumab, have been associated with adverse effects on cardiac function [7]. The chemotherapy-induced left ventricular (LV) dysfunction is not uncommon and may lead to premature cessation of anticancer therapy with obvious sequalae on prognosis. Consequently, early cardiac assessment and regular cardiac monitoring is required in BC patients receiving cardiotoxic therapy [8]. Resting echocardiography is the main imaging modality to detect cancer therapy-related cardiovascular dysfunction (CTRCD). It is broadly available, radiation free, safe, and cost-effective with capacity for repeated measurements [9].

Left ventricular ejection fraction (LVEF) remains one of the main criteria for diagnosing cancer therapy-related cardiac dysfunction (CTRCD). However, a decline in LVEF typically occurs at a later stage of CTRCD, when the damage may already be irreversible [10,11]. Subclinical cardiac dysfunction is more common in chemotherapy-treated patients, and can be detectable through novel echocardiographic techniques, especially speckle tracking analysis (STA) [12]. Its main index-global longitudinal strain (GLS) may identify myocardial dysfunction before it becomes apparent with LVEF decline or symptoms onset [5,7,8]. Hence, GLS has been advocated for early detection of CTRCD and international guidelines strongly recommend its calculation at baseline and during chemotherapy at regular intervals to determine chemotherapy continuation and guide timely interventions [13,14,15]. One of the main drawbacks of GLS is the high influence of blood pressure on its levels. Myocardial work (MW) has been recently introduced in clinical studies as an assay of LV strain–pressure loop. It integrates strain measurements with pressure, offering a more objective load-adjusted assessment of LV function, and concomitantly addresses key limitations of LVEF and GLS. The Global Work Index (GWI) reflects total myocardial work, Global Constructive Work (GCW) represents effective contraction, Global Wasted Work (GWW) indicates inefficient or wasted effort, and Global Work Efficiency (GWE) measures the proportion of effective to total work [16]. From a theoretical perspective, those indices could enhance early detection of subclinical cardiac dysfunction, by providing insights into both global and segmental myocardial energetics [17,18]. There are scarce data on the diagnostic accuracy and prognostic value of MW in cardio-oncology.

Emerging clinical studies indicate that chemotherapy-treated patients may develop diastolic dysfunction during their cancer therapy [19,20]. Isolated diastolic dysfunction has been associated with poor clinical outcomes due to its link with heart failure with preserved ejection fraction (HFpEF). Conventional resting measures of advanced diastolic function are predictive of future events [21]. However, when diastolic dysfunction is less advanced (grade I or II) and HFpEF diagnosis is suspected, diastolic stress echocardiography (DSTE) is recommended in combination with resting echocardiography and natriuretic peptides [22]. HFA-PEFF score, a diagnostic algorithm developed by the Heart Failure Association of the European Society of Cardiology for the diagnosis of HFpEF, may differentiate cardiac from non-cardiac origin of HF symptoms in individuals with preserved LVEF. In this case, the added value of DSTE has been outlined by both the American Society of Echocardiography and the European Association of Cardiovascular Imaging [21]. Despite its potentiality, the role of DSTE in cardio-oncology remains underexplored [10,23].

Among the most clinically relevant circulating biomarkers are B-type natriuretic peptide (BNP) depicting ventricular wall stress and cardiac troponin (cTn) reflecting myocardial injury. The elevation of their levels after chemotherapy onset, particularly with anthracyclines or HER2-targeted agents, are strongly associated with the development of LV dysfunction and have been recommended to guide prompt cardioprotective strategies [24]. They are particularly useful for identifying early myocardial injury due to HF. Galectin-3, a marker of fibrosis and cardiac remodeling, is emerging as a potential tool for long-term risk stratification in CTRCD, though its routine use in oncology remains uninvestigated [25]. A significant proportion of BC patients undergoing chemotherapy complain of general symptoms of exertional dyspnea and/or fatigue in the absence of other obvious reasons (e.g., anemia); it is therefore almost impossible to distinguish them from the clinical manifestations of CTRCD. We hypothesized that advanced echocardiographic techniques (GLS, MW indices, DSTE) combined with cardiac biomarkers could improve early detection of CTRCD and identify a HFpEF phenotype in breast cancer patients undergoing chemotherapy, beyond conventional LVEF monitoring. In addition, we assessed the potential role of diastolic stress echocardiography (DSTE) in identifying patients with an HFpEF phenotype within this cohort.

## 2. Methodology

### 2.1. Study Design and Participants

This is a prospective case–control study. We enrolled female patients with BC, eligible for anticancer chemotherapy with anthracyclines and/or trastuzumab (chemotherapy group-CTG, *n* = 145). All cancer patients were recruited consecutively from the outpatient clinics of the Bank of Cyprus Oncology Centre and were followed up for 12 months. A second group of age-matched healthy female volunteers (2:1 ratio, *n* = 65) served as the control group (CG) at baseline. Most of controls were relatives, friends, or siblings of recruited cancer patients, invited to participate voluntarily. They were free from any chronic disease and were not on any chronic regimen. This study was approved by the National Bioethics Committee of Cyprus (EEBK/EEP/2021/34) and conducted in accordance with the Declaration of Helsinki and all participants provided signed informed consent.

We excluded patients fulfilling any of the following criteria: known HF, coronary artery disease (CAD), primary valvular disease of at least moderate severity, inherited or acquired cardiomyopathy, atrial fibrillation, inability to perform an exercise test, or inadequate acoustic windows on echocardiography. To exclude CAD, all candidates underwent a functional test for myocardial ischemia or coronary computed tomography angiography prior to study entry.

At baseline, all participants underwent a comprehensive assessment before anticancer therapy initiation. This included medical history and demographic data collection (comorbidities, medications), physical examination (body mass index, blood pressure, heart rate), resting electrocardiogram, and echocardiographic assessments—including resting echocardiography and DSTE. Blood samples were collected for the analysis of cTn, NT-proBNP and galectin-3. Echocardiographic procedures are described in detail in subsequent sections. Follow-up evaluations were conducted every 3 months up to the end of the 12-month study period. The baseline assessment was repeated at each visit including blood assays. Based on current international guidelines, the chemotherapy group was further subdivited into subgroups of patients developing CTRCD and CTRCD-free subgroup. CTRCD was defined according to current ESC guideline criteria as an absolute reduction in LVEF ≥ 10% to a value <50% and/or a relative reduction in GLS > 15% from baseline, with supportive evidence from biomarker elevation (troponin, NT-proBNP) when available. Furthermore, we calculated the HFA-PEFF score, incorporating functional (E/e′, TRVmax, GLS), morphological (LAVI, LV mass index, relative wall thickness), and biomarker (NT-proBNP) domains. Major and minor criteria were weighted 2 and 1 point, respectively. Scores ≥ 5 indicated HFpEF, 2–4 intermediate probability, and ≤1 excluded HFpEF.

### 2.2. Echocardiography

All echocardiography tests (resting and DSTE) were performed at the Medical School University of Cyprus by a single experienced operator at baseline and during every scheduled visit. Advanced analysis of already acquired echocardiographic images was performed offline, using specific software (Echopac version 203, Chicago, IL, USA) and the interpretation of the results was performed by two cardiologists blinded to patients’ history and symptoms. At rest, the following parameters were assessed: left ventricular ejection fraction (LVEF, biplane Simpson method), global longitudinal strain (GLS), myocardial work indices [Global Work Index (GWI), Global Constructive Work (GCW), Global Wasted Work (GWW), Global Work Efficiency (GWE)], diastolic function indices (E, A, E/A ratio, E/E′), left atrial volume index (LAVI), and right ventricular systolic pressure (RVSP) estimated from tricuspid regurgitation velocity (TRVmax). For DSTE we used an ergocycle (Ergoline ergoselect 12, Bitz, Germany). The DSTE was performed in semi-supine position according to a well-established protocol (WHO) which has been officially proposed [26]. The initial workload was adapted from 25 to 75 W, based on patient’s physical capacity. Parameters reassessed under stress included transmitral flow velocities (E, A), mitral annular tissue Doppler velocities (E′, A′), E/E′ ratio, TRVmax (for estimation of pulmonary artery systolic pressure), and LVEF. The test was ended when E and A trans-mitral waves were fused. All performed tests were technically adequate with acquisition of optimal views.

### 2.3. Blood Assays

Blood samples were collected after overnight fasting and were centrifugated. The remaining serum was stored in a deep freezer at −80 °C. The measurements of serum high-sensitivity cardiac troponin I (hs-cTnI) and NT-proBNP were conducted using the Alinity analyzer from Abbott Diagnostics (Abbott Park, Libertyville Township, IL, USA). Serum concentrations of Galectin-3 were determined using a commercially available enzyme-linked immunosorbent assay (ELISA) kit (FineTest^®^, EH0145, Wuhan Fine Biotech Co., Ltd., Wuhan, China), following the manufacturer’s instructions. Serum samples (50 μL per well) were incubated with biotin-labeled detection antibody and horseradish peroxidase (HRP)-conjugated streptavidin. The signal was developed using a tetramethylbenzidine (TMB) substrate and quantified spectrophotometrically at 450 nm. All serum samples were processed and stored at −80 °C until analysis. Before measurement, samples were equilibrated to room temperature and diluted 1:10 with sample dilution buffer. The assay sensitivity was 0.094 ng/mL, with a working range of 0.156–10 ng/mL. The intra-assay coefficient of variation (CV) ranged from 4.23% to 5.22%, and the inter-assay CV ranged from 5.02% to 5.43%, based on internal quality control samples with low, medium, and high galectin-3 concentrations. All measurements were performed in duplicate, and standard curves were constructed.

### 2.4. Statistical Analysis

All continuous variables were tested for normality using the Shapiro–Wilk test. Normally distributed data are presented as the mean ± standard deviation (SD), while non-normally distributed variables are expressed as median (interquartile range). Categorical variables are presented as frequencies (percentages). Comparisons between groups were performed using independent-samples *t*-test for continuous variables, and chi-square test for categorical data. Within-group changes over time were assessed using paired-samples *t*-tests. Pearson or Spearman correlation analyses were used to explore associations among echocardiographic parameters, biomarkers, and clinical features, as appropriate. Receiver Operating Characteristic (ROC) curve analysis was used to evaluate the diagnostic performance of selected variables, with AUC, sensitivity, specificity, and optimal cut-off values reported. Multivariate logistic regression analysis was performed to identify independent predictors of CTRCD, including only variables with *p* < 0.05 in univariate analysis. To minimize the risk of overfitting given the limited number of CTRCD cases (*n* = 28), only variables with significant univariate associations (*p* < 0.05) were included in multivariate logistic regression, maintaining an acceptable events-per-variable ratio. As this was an exploratory, hypothesis-generating study, no formal a priori sample size calculation was performed. Interobserver variability was assessed by reanalyzing 10% of echocardiographic images by two independent blinded observers. A two-tailed *p*-value < 0.05 was considered statistically significant. Statistical analyses were performed using SPSS software (version 25.0; IBM Corp., Armonk, NY, USA).

## 3. Results

### 3.1. Baseline Participant Characteristics

We initially enrolled 145 women with newly diagnosed breast cancer treated with anthracycline-based chemotherapy with or without anti-HER2 therapy (trastuzumab +/− pertuzumab). A total of133 patients completed this study, and full data were obtained for statistical analysis. The control group consisted of 65 age-matched healthy women (roughly 2:1 ratio). At baseline, a proportion of oncologic patients had diabetes (7.5%), hypertension (13.5%), dyslipidemia (15%) already treated with medications. At the beginning of this study, a comparable number of oncologic patients and healthy individuals were active smokers (22.6% vs. 20%). There were no significant differences between groups across hemodynamic parameters, echocardiographic findings [either classical (E/A, E/E′, TRVmax) or novel (GLS, myocardial work indices)], plasma levels of biomarkers (NT-proBNP, troponin and galectin-3) and DSTE parameters (*p* > 0.05; Table 1).

### 3.2. Follow-Up

At the end of this study, 133 patients completed the 12-month follow-up. In the meantime, 12 patients were excluded from analysis: 4 patients died due to cancer, 2 cancer patients lost at least 2 of their regular visits, and 6 patients wanted to withdraw from this study for personal reasons. All recruited oncologic patients complained about fatigue and most of them for exertional dyspnea during follow-up. Those symptoms raised the question whether they have developed CTRCD or not. Based on current international diagnostic criteria, our study population was subdivided into two subgroups: CTRCD subgroup, (*n* = 28) that is patients who developed CTRCD at any time of follow-up; and CTRCD-free subgroup, (*n* = 105) patients who did not develop CTRCD within 12 months from chemotherapy onset. At baseline, those subgroups did not differ across all demographics, echocardiographic and hemodynamic parameters. During the follow-up period, 2 women were hospitalized due to acute heart failure and no other acute cardiovascular events were reported. Patients with CTRCD were treated with any combination of b-blockers, ACE inhibitors and diuretics when tolerated. In addition, the chemotherapy regimen was modified in that subgroup: chemotherapy dosage reduction, prolongation of intervals between chemotherapy sessions, temporary cessation of chemotherapy or change in chemotherapeutic agents. Notably, none of the CTRCD patients permanently stopped chemotherapy. Regarding the rest of clinical parameters, smoking rate was remarkably decreased in the cancer group during follow-up after oral instruction for smoking cessation. Anti-hypertensive medications were slightly modified in some women (dose reduction), but the total number of hypertensive women remained unaltered (Table 2).

#### 3.2.1. Follow-Up of Echocardiography Indices

Compared to baseline values, the CTRCD subgroup presented a considerable increase in GLS (from −21.22 (1.57) to −16.69 (1.69)%, *p* < 0.001) and a remarkable decline in LVEF levels (from 61.00 (4.00)% to 54.00 (4.00)%, *p* < 0.001), as expected. Hence, at the end of this study, the CTRCD subgroup had lower LVEF and higher GLS and NT-proBNP levels than CTRCD-free and healthy controls (*p* < 0.001). Notably, only 6 patients out of 28 in CTRCD subgroup had LVEF < 50% and were excluded from repeated DSTE till the end of this study. Concerning the indices of systolic function of RV both TAPSE and S’ become smaller in patients with CTRCD than those without CTRCD, but without achieving significant level (Table 3).

MW parameters did not demonstrate notable differences between groups at baseline (*p* > 0.05). The Global Work Index (GWI) and the Global constructive work (GCW) were significantly reduced in CTRCD compared to CTRCD-free subgroup (*p* = 0.042, *p* = 0.023, respectively), suggesting reduced myocardial performance. Global Wasted Work (GWW) was slightly higher in the CTRCD group, though this difference was not statistically significant. Global Work Efficiency (GWE) was lower in the CTRCD patients than CTRCD-free counterparts (*p* = 0.028). Table 2 and Table 3 summarize the measurements of CRTCD vs. CRTCD-free patients at baseline and the end of this study.

#### 3.2.2. Follow-Up of Biomarkers

The CTRCD subgroup showed a significant increase in NT-proBNP levels (*p* < 0.001) during follow-up, compared to the CTRCD-free group and healthy controls. Troponin levels remained within normal range during follow-up, but they were markedly elevated within the CTRCD subgroup compared to CTRCD-free and healthy controls, indicating a subclinical myocardial injury. No significant changes in galectin-3 levels were noted in both subgroups, limiting its contribution to detection of CTRCD (*p* > 0.05) (Table 2).

#### 3.2.3. Follow-Up and DSTE Results

ESE-derived TRVmax post-chemotherapy was significantly higher in the CTRCD group compared to the CTRCD-free group, suggesting elevated pulmonary pressures presumably due to higher LV filling pressures under stress conditions in patients with cardiac impairment. That was also implicated by the post-chemotherapy elevation in E/A and E/E′ ratio in the CTRCD versus CTRCD-free subgroup, more reaching statistically significant difference (*p* < 0.001).

Based on resting echocardiography, NT-proBNP levels and DSTE results, we calculated the HFA-PEFF score. Sixteen patients without fulfilling the criteria for CTRCD, appeared to have high probability of HFpEF. In particular, 13 patients achieved high HFA-PEFF score = 4 (DSTE contribution included), while another 3 patients achieved HFA-PEFF score ≥ 5 and were labeled as HFpEF patients. We decided to compare the subgroup of patients with highly suspected or diagnosed HFpEF (HFA-PEFF score ≥ 4) to the CTRCD group (Table 4). As expected the CTRCD subgroup had worse systolic function (lower LVEF and increased GLS) compared to the HFpEF subgroup. The former group had even lower GCW (*p* = 0.012), GWE (*p* = 0.019) and a trend for lower GWI (*p* = 0.070), During DSTE, those two subgroups did not differ in the main parameters such as E/A ratio, E/E′ and TRVmax. The CTRCD group maintained markedly higher NT-proBNP and troponin levels than their counterparts with suspected or established HFpEF (*p* < 0.001).

#### 3.2.4. Subgroup Analysis Excluding Baseline Comorbidities

To minimize the influence of baseline cardiovascular risk factors, we performed a subgroup analysis excluding patients with diabetes or hypertension. In this subgroup, GLS impairment and NT-proBNP elevation remained significant predictors of CTRCD, while myocardial work indices (GCW, GWI) also showed predictive value. Troponin demonstrated a non-significant trend toward higher levels in the CTRCD group. These findings suggest that advanced echocardiographic indices and biomarkers predict CTRCD independently of common baseline comorbidities (Table 5).

### 3.3. Correlations

Univariate analysis revealed that CTRCD diagnosis was significantly associated with multiple indicators of cardiac dysfunction like NT-proBNP, troponin, DSTE-related E/E′, and DSTE-related TRVmax. It was also negatively correlated with key functional and myocardial performance indices, including LVEF, GLS, GWI, GCW, and GWE—supporting the notion that indices of subtle myocardial dysfunction are closely linked to CTRCD development. To further explore those associations, logistic regression analysis was conducted and NT-proBNP, troponin, GLS, and DSTE-related TRVmax remained independent predictors of CTRCD development in our model. Together, these findings emphasize the diagnostic and prognostic value of combining biomarkers assay with advanced echocardiographic techniques (particularly DSTE-related TRVmax and strain imaging) for early risk stratification and detection of CTRCD.

Since DSTE-related TRVmax was highly associated with CTRCD, we tested its discriminatory ability in the whole population undergoing chemotherapy. The area under the curve (AUC) was 0.873 (95% CI: 0.812–0.935, *p* < 0.001), indicating strong diagnostic performance. The optimal cut-off value based on the coordinates of the curve appears to be around 2.79 m/s, which provides a sensitivity of 89% and a specificity of 76%, suggesting an effective detection of all true positive cases while maintaining reasonable specificity (Table 6) (Figure 1). These findings support the utility of DSTE-related TRVmax as a non-invasive stress echocardiographic marker for early identification of CTRCD or cardiac stiffening associated with high LV filling pressure or HFpEF in patients with preserved LVEF.

## 4. Discussion

This study investigated the diagnostic value of advanced echocardiographic techniques, specifically GLS, MW indices and DSTE, for early detection of CTRCD among women with BC undergoing cardiotoxic therapy. Our main findings included the significant deterioration of classical indices, such as LVEF, GLS, and NT-proBNP in CTRCD vs. CTRCD-free patients and healthy controls. In addition those expected results, the presence of CTRCD was associated with worse values of GWI, GCW, GWE and DSTE-TRVmax than those not affected by chemotherapy. In parallel to NT-proBNP, hs-cTnI elevated along with CTRCD development, but remained within normal range. Galectin-3 failed to add any value in CTRCD diagnosis. In logistic regression analysis, NT-proBNP, Troponin, GLS, and DSTE-related TRVmax remained independent predictors of CTRCD development. Finally, an elevated DSTE-related TRVmax had the potential to identify patients with high HFA-PEFF score (≥4) among CTRCD-free patients. Our model using advanced echocardiographic markers, cardiac biomarkers and DSTE may be able to detect early, subclinical myocardial damage caused by anthracycline-based chemotherapy and anti-HER2 therapies [27,28].

Baseline characteristics were well matched between the chemotherapy and control groups, with no significant baseline differences in echocardiographic parameters or biomarkers, supporting the appropriateness of the selected study population. However, during the follow-up period, clear distinctions emerged between patients who developed CTRCD and those who remained CTRCD free. Consistent with existing literature, significant reduction in LVEF and increased GLS were observed among patients who subsequently developed CTRCD. Only a few patients presented with LVEF < 50%, highlighting GLS’s sensitivity in detecting early myocardial deformation preceding overt systolic dysfunction. These results align with prior studies advocating for serial GLS monitoring during chemotherapy to enhance early risk stratification [29,30].

Further enhancing diagnostic precision, MW analysis demonstrated lower values in GWI, GCW, and GWE in the CTRCD subgroup at follow-up compared to baseline values and those of other groups (CTRCD-free and healthy controls subjects). These MW indices, integrating strain data with myocardial afterload and pressure data provide a more comprehensive assessment of cardiac function compared to traditional strain parameters alone [31]. This underscores the potential clinical utility of incorporating MW indices into routine echocardiographic surveillance for patients undergoing cardiotoxic chemotherapy. In this study, patients who developed CTRCD showed significantly reduced GWI, GCW, and GWE, indicating impaired myocardial performance. These indices provided superior, load-independent diagnostic value over traditional measures like LVEF and GLS, especially in detecting subclinical dysfunction. Prognostically, impaired MW has been associated with elevated cardiac biomarkers and symptomatic deterioration, suggesting that MW indices can help identify patients at higher risk for long-term cardiac events; making them valuable tools for both early detection and risk stratification in cardio-oncology [32].

Cardiac biomarkers (NT-proBNP and troponin) significantly complemented imaging findings. Elevated NT-proBNP levels indicated increased myocardial wall stress and symptomatic heart failure tendencies, while elevated troponin reflected subclinical myocardial injury. The 4-fold increase in NT-proBNP levels in the CTRCD subgroup and 2-fold increase in the subgroup with diagnosed or high probability for HFpEF, outline the clinical relevance of this biomarker. On the other hand, the rise in the circulating levels of troponin implicated the cardiotoxic effects of chemotherapy; however, those levels remained within normal range. Therefore, it should be further investigated whether the fluctuation of otherwise “normal” levels of troponin reflect unambiguously cardiotoxicity. Both biomarkers, together with imaging modalities, could enhance the predictive accuracy for CTRCD, emphasizing the need for multimodal cardiac monitoring as recommended in current ESC cardio-oncology guidelines [13]. On the other hand, we failed to identify any significant difference between groups, regarding galectin-3 levels. This is an important factor of myocardial fibrosis, and it is associated with cardiac-related mortality [33]. However, it has shown higher accuracy in HF with reduced LVEF rather than among patients with preserved LVEF [34]. This may explain the absence of significant change in its levels in the CTG versus CG.

In our study, we hypothesized that the implementation of DSTE would substantially enhance the sensitivity of HF diagnosis. Indeed, patients with CTRCD and preserved LVEF appeared during DSTE with a substantial increase in TRVmax and E/e′ ratio. Remarkably, DSTE-derived TRVmax exhibited exceptional diagnostic performance, with an area under the curve (AUC) of 0.873, effectively discriminating patients with CTRCD and HF-related symptoms from CTRCD-free counterparts. complaining about symptoms unrelated to cardiac function as consequence of their disease and/or chemotherapy. This outcome is clinically significant as it introduces DSTE-derived TRVmax as a non-invasive, reliable echocardiographic biomarker capable of timely recognition of CTRCD. Moreover, the development of diastolic dysfunction predominantly precedes systolic cardiac dysfunction [35]. The ability of non-invasive diastolic stress testing to successfully unmask diastolic dysfunction and distinguish the cardiac from non-cardiac origin of dyspnea and fatigue, highlights its potential application in cardio-oncology. Moreover, it is a test with low workload and low intensity (up to 100–110 bpm) feasible for cancer patients with exaggerated fatigue and comorbidities [33].

Additionally, we detected within the CTRCD-free group, a number of patients characterized by exertional dyspnea during the follow-up period and with a considerable elevation in TRVmax and E/e′ ratio during DSTE. That subgroup achieved a high HFA-PEFF score implicating a high probability for HFpEF. Most accumulated points predominantly derived by the increased values of GLS, NT-proBNP (major criteria) and the DSTE-related TRVmax. suggesting pronounced diastolic impairment and elevated filling pressure. The identification of patients with high HFA-PEFF score, not yet fulfilling traditional CTRCD criteria, further emphasizes the importance of nuanced evaluation for subclinical cardiac dysfunction and especially diastolic dysfunction. Such stratification enables the timely initiation of cardioprotective therapies, potentially preventing progression to HFrEF and symptoms [35]. We and other researchers have hypothesized that chemotherapy may affect the diastolic cardiac function, at early stage, causing HF symptoms, without necessarily leading to classic CTRCD. The available evidence is weak and limited and more studies are needed to investigate the interplay between chemotherapeutic agents, cardiac stiffening and diastolic dysfunction [36]. DSTE has emerged as a complementary tool for identifying diastolic dysfunction in patients complaining about unexplained dyspnea on exertion [37,38]. Consequently, this approach is now endorsed by both the American Society of Echocardiography and the European Association of Cardiovascular Imaging [39]. Overall, the role of DSTE has been upgraded, since there is a growing body of evidence supporting the use of diastolic stress testing across the HF spectrum. The proposed cut-off values of E/e′ and TRVmax should be re-considered, since some investigators have proposed alternative values or parameters (sharp elevation in TRVmax at early stages of DSTE, etc.) [38].

Our findings support the integration of advanced echocardiographic and biomarker parameters into longitudinal cardio-oncology care. Current ESC guidelines recommend that a relative reduction in GLS > 15% from baseline, or persistent elevations in NT-proBNP and troponin, should prompt intensified surveillance and consideration of early cardioprotective therapy. In this context, our results reinforce the clinical value of serial GLS and biomarker monitoring during chemotherapy. Moreover, the novel observation that DSTE-derived TRVmax independently predicts CTRCD suggests that stress echocardiography could help unmask subclinical diastolic dysfunction and guide management in patients with unexplained symptoms. Although our study was not designed to establish definitive therapeutic thresholds, these parameters may support individualized treatment decisions and timely initiation of cardioprotective interventions, thereby minimizing interruptions of oncologic therapy.

Clinical implications of our study advocate a new approach of oncological patients clustering advanced echocardiographic techniques (STA and DSTE), MW parameters, and cardiac biomarkers to formulate a new grading scale for CTRCD and integrate HFpEF investigation in diagnostic algorithms. An early identification of cardiac dysfunction in cancer patients could allow proactive therapeutic interventions, thus potentially reducing therapy interruptions and preserving long-term cardiac health. In our study, CTRCD patients were treated according to current recommendations with beta-blockers, ACE inhibitors and furosemide. Several pharmacological strategies have demonstrated efficacy in reducing cardiotoxicity. Dexrazoxane, an iron-chelating agent, reduces anthracycline-induced oxidative injury and DNA damage [40]. Neurohormonal antagonists (beta-blockers, ACE inhibitors, and ARNI) have shown benefits in preserving myocardial strain and LVEF during chemotherapy [41]. Statins may provide pleiotropic anti-inflammatory and antioxidant effects, mitigating anthracycline- and trastuzumab-associated injury [42].

With cardioprotective agents careful monitoring is required due to potential interactions. Dexrazoxane may alter anthracycline pharmacokinetics and potentially reduce anticancer efficacy if used early [43], while ACE inhibitors/ARNIs may increase the risk of hypotension or renal dysfunction when used with VEGF inhibitors [44]. Statins, metabolized by CYP3A4, may interact with tyrosine kinase inhibitors, raising the risk of myopathy or liver toxicity [45]. In addition, many newer cancer therapies can prolong QT interval or elevate blood pressure, requiring caution when used alongside antiarrhythmics [46]. These interactions highlight the importance of individualized therapy. Incorporating cardiovascular biomarkers and advanced imaging into early-phase oncology trials may help identify safer dose regimens and guide patient selection for novel therapies. This integrative approach not only enhances the ability to detect subclinical cardiac injury at an early stage but also supports the development of personalized treatment strategies that balance oncologic efficacy with cardiovascular safety.

There are several limitations in the present study. First, although the sample size was adequate to demonstrate significant results for most of our study objectives, the relatively small number of patients who developed CTRCD (*n* = 28) limits statistical power for regression and ROC analyses, and some secondary analyses may therefore be underpowered. Moreover, no formal a priori sample size calculation was performed. Second, the relatively short 12 month follow-up duration may not capture long-term cardiovascular outcomes or late-onset cardiotoxicity. Longer follow-up is necessary to assess whether early subclinical changes evolve into overt HF or if cardioprotective interventions can reverse them. Secondly, we excluded patients with suboptimal echocardiographic image quality—due to obesity or breast surgery. This introduces potential selection bias, as these individuals may represent a higher-risk group for CTRCD and are frequently encountered in real-world oncology practice. Importantly, the study cohort included patients who received various types of anticancer therapies with differing cardiotoxic potential. While doxorubicin is a well-known agent associated with CTRCD, not all patients received this drug. In our study, all women with breast cancer received anthracycline-based chemotherapy (FEC, AC, or EC), with or without Paclitaxel/Docetaxel, and in some cases combined with anti-HER2 therapy (trastuzumab ± pertuzumab). This reflects a small heterogeneity since all women received anthracyclines and a small proportion of them (18%) additional agents used in sequence or combination. The cardiotoxic risk varies substantially across these agents, and future large-scale studies are warranted to evaluate their specific and cumulative impact on cardiac outcomes. Although GLS and DSTE demonstrated strong diagnostic performance in our study, their reproducibility in clinical practice is subject to operator experience, image quality, and vendor-related variability. In our study, the use of a single experienced operator reduced intra-study variability, but this does not reflect real-world, multi-center conditions. We acknowledge the absence of a formal inter-observer variability assessment as a limitation. Finally, the single-center design and absence of external validation limit the generalizability of our results. Cut-off values for E/e; and TRVmax, should be addressed in future research through longitudinal multicenter trials in order to refine diagnostic strategies.

## 5. Conclusions

This study highlights the emerging role of advanced echocardiographic techniques—MW indices and DSTE—in the early detection of myocardial systolic and diastolic dysfunction in women undergoing chemotherapy for BC. Traditional reliance on LVEF alone frequently results in delayed diagnosis, missing the early subclinical phase when cardiotoxicity is potentially reversible. Combined with cardiac biomarkers like NT-proBNP and troponin, those novel imaging modalities offer a powerful, multimodal approach for comprehensive cardiac surveillance. Notably, DSTE emerged as a valuable tool for unmasking exertional dyspnea and/or fatigue related to diastolic dysfunction and distinguishing true cardiac impairment from common, nonspecific chemotherapy-related symptoms. While our findings support the potential role of DSTE and STA indices in early detection, recommendations for routine clinical implementation are premature. Future large-scale, prospective studies are essential to validate these findings, establish standardized thresholds, and guide the implementation of advanced echocardiographic surveillance as part of evidence-based cardio-oncology protocols.

## Figures and Tables

**Figure 1 biomedicines-13-02341-f001:**
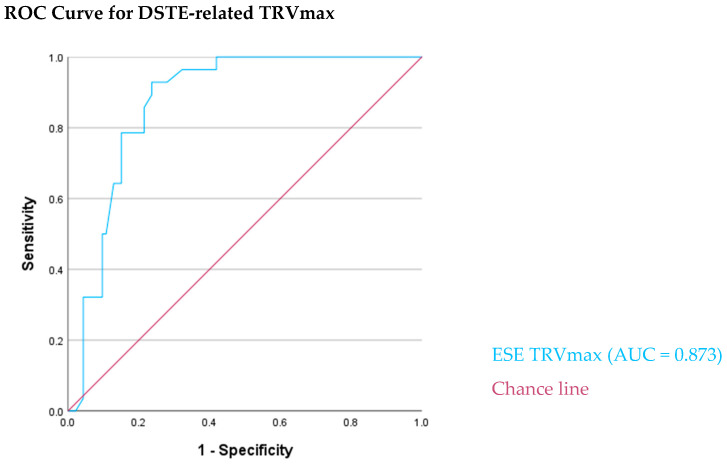
Demonstrated excellent discriminatory ability of DSTE-related TRVmax in predicting chemotherapy-related cardiac dysfunction (CTRCD). The area under the curve (AUC) was 0.873 (95% CI: 0.812–0.935, *p* < 0.001), indicating strong diagnostic performance.

**Table 1 biomedicines-13-02341-t001:** Comparison of Clinical and Echocardiographic Parameters Between Cancer Patients and Healthy Controls at Baseline.

Variable	Cancer Patients (*n* = 133)	Healthy Controls (*n* = 65)	*p*-Value
Age	52.00 (12.00)	51.00 (6.00)	0.263
SBP (mmHg)	133.00 (22.00)	130.00 (12.00)	0.216
DBP (mmHg)	83.00 (10.00)	82.00 (7.00)	0.574
LVEF (%)	66.00 (5.00)	67.00 (7.00)	0.761
E/E′	7.19 (2.17)	7.06 (1.50)	0.854
TAPSE (cm)	2.50 (0.30)	2.60 (0.40)	0.816
RVS (cm/s)	14.20 (0.40)	14.70 (0.20)	0.768
TRVmax (m/s)	2.25 (0.23)	2.58 (0.25)	0.277
LAVI (mL/m^2^)	30.39 (8.13)	28.91 (6.16)	0.327
GLS (%)	−21.16 (2.05)	−21.17 (1.57)	0.991
GWI (mmHg%)	2223.31 (503.75)	2134.79 (292.77)	0.652
GWW (mmHg%)	115.79 (50.89)	109.47 (40.97)	0.407
GCW (mmHg%)	2505.94 (470.53)	2383.12 (334.93)	0.602
GWE (%)	93.91 (3.78)	94.62 (3.28)	0.620
Troponin (ng/L)	2.36 (1.49)	2.13 (0.90)	0.573
NT-proBNP (pg/mL)	45.19 (16.41)	43.93 (22.46)	0.589
Galectin-3 (ng/mL)	13.15 (6.32)	12.49 (12.58)	0.326
ESE E/E′	6.74 (1.23)	6.18 (0.93)	0.707
ESE TRVmax (m/s)	2.30 (0.37)	2.22 (0.34)	0.801

Continuous variables are presented as the mean (SD); *p*-values are derived from independent-samples *t*-tests.

**Table 2 biomedicines-13-02341-t002:** Baseline and Follow-Up Demographics and Biomarker levels Between Groups With CTRCD and Without CTRCD.

Variables	CTRCD Group (*n* = 28)		CTRCD-Free Group (*n* = 105)		P1	P2
	**Baseline**	**End**	**Baseline**	**End**		
Age	54.00 (13.00)		52.00 (11.00)	N/A	0.223	N/A
BMI (kg/m^2^)	25.76 (5.31)	25.11 (4.66)	27.87 (7.18)	26.98 (4.22)	0.056	0.046
SBP (mmHg)	136.00 (16.00)	137.00 (17.00)	132.00 (24.00)	133.00 (23.00)	0.106	0.148
DBP (mmHg)	86.00 (10.00)	85.00 (9.00)	82.00 (10.00)	83.00 (10.00)	0.151	0.096
Diabetes (*n*)	3	3	7	7	0.301	0.301
Hypertension (*n*)	6	6	12	12	0.812	0.812
Dyslipidemia (*n*)	5	5	15	15	0.222	0.222
Smoking (*n*)	12	4	18	6	0.456	0.895
NT-proBNP (pg/mL)	50.37 (20.02)	200.06 (113.61)	44.62 (16.71)	61.49 (32.27)	0.138	<0.001
Troponin (ng/L)	2.56 (1.21)	12.42 (2.73)	2.29 (1.55)	3.95 (2.65)	0.170	<0.001
Galectin-3 (ng/mL)	13.23 (4.50)	13.27 (5.74)	13.12 (6.83)	12.56 (3.88)	0.466	0.081

Continuous variables are presented as the mean (SD); Categorical variables are presented as *n*. *p*-values are derived from independent-samples *t*-tests. P1, *p*-value of differences between groups at baseline. P2, *p*-value of differences between groups at the end. *n*, number of patients.

**Table 3 biomedicines-13-02341-t003:** Baseline and Follow-up Echocardiographic Parameters Between Groups with CTRCD and Without CTRCD.

Variables	CTRCD Group (*n* = 28)		CTRCD-Free Group (*n* = 105)		P1	P2
	**Baseline**	**End**	**Baseline**	**End**		
LVEF (%)	61.00 (4.00)	54.00 (5.00)	63.00 (6.00)	63.00 (7.00)	<0.001	<0.001
GLS (%)	−21.22 (1.96)	−16.68 (1.55)	−21.14 (2.08)	−20.31 (1.82)	0.759	<0.001
GWI (mmHg%)	2375.64 (477.00)	1939.92 (415.01)	2055.89 (600.84)	2340.14 (364.35)	0.012	<0.001
GCW (mmHg%)	2575.70 (422.00)	N/A	2409.95 (580.23)	N/A	0.163	N/A
GWW (mmHg%)	138.50 (106.44)	83.55 (59.15)	134.25 (129.15)	72.85 (31.53)	0.881	<0.091
GWE (%)	94.30 (3.31)	95.56 (2.40)	93.80 (3.91)	96.42 (1.22)	0.574	<0.001
TAPSE (cm)	2.40 (0.40)	2.30 (0.40)	2.70 (0.40)	2.40 (0.30)	0.567	0.808
TV S′ (cm/s)	12.40 (2.10)	11.50 (1.60)	12.70 (1.80)	12.00 (2.50)	0.775	0.552
E/E′	7.75 (3.14)	6.89 (1.86)	7.02 (1.77)	7.66 (2.52)	0.133	0.053
E/A	0.99 (0.31)	0.95 (0.22)	1.00 (0.34)	0.99 (0.30)	0.973	0.527
DSTE TRVmax (m/s)	2.33 (0.44)	3.05 (0.46)	2.20 (0.46)	2.31 (0.44)	0.801	<0.001

Notes: P1 = between-group comparison at baseline; P2 = between-group comparison at study end. Continuous variables are presented as the mean (SD).

**Table 4 biomedicines-13-02341-t004:** Comparison of Biomarker and Echocardiographic Parameters Between Patients With Suspected or Established HFpEF and Those With CTRCD at Follow-up.

Parameter	CTRCD Group (*n* = 28)	Suspected or Established HFpEF Group (*n* = 16)	*p*-Value
GLS (%)	−16.72 (1.61)	−19.10 (1.77)	<0.001
NT-proBNP (pg/mL)	199.53 (118.05)	113.13 (51.70)	<0.001
Troponin (ng/L)	12.59 (2.77)	7.10 (3.30)	<0.001
LVEF (%)	53.00 (11.00)	61.63 (5.58)	<0.001
TRVmax baseline (m/s)	2.10 (0.19)	2.36 (0.21)	0.004
ESE-derived TRVmax (m/s)	3.05 (0.27)	3.18 (0.24)	0.049

Continuous variables are presented as the mean (SD).

**Table 5 biomedicines-13-02341-t005:** Subgroup Analysis of Predictors of CTRCD in Breast Cancer Patients Without Baseline Comorbidities.

Variable	CTRCD Group (*n* = 19)	CTRCD-Free Group (*n* = 86)	*p*-Value
GLS (%)	−16.20 (2.10)	−20.40 (2.50)	<0.001
NT-proBNP (pg/mL)	198.50 (110.30)	59.70 (31.80)	<0.001
Troponin (ng/L)	12.10 (2.80)	3.90 (2.60)	0.080
GCW (mmHg%)	1650.00 (350.00)	2150.00 (420.00)	0.014
GWI (mmHg%)	1670.00 (310.00)	2280.00 (450.00)	0.010

**Table 6 biomedicines-13-02341-t006:** Logistic Regression Analysis for Predictors of CTRCD. Wald χ^2^ Statistics and *p*-Values Are Presented. Variables With *p* < 0.05 Were Considered Significant Predictors.

Variable	Wald χ^2^	*p*-Value
ESE E/E′	0.215	0.643
ESE TRVmax	17.590	<0.001
ESE E/A ratio	0.875	0.350
Troponin	39.007	<0.001
NT-proBNP	40.050	<0.001
LVEF	5.365	0.021
GLS	22.416	<0.001
GWI	10.970	<0.001
GCW	8.601	0.003
GWE	1.515	0.218

## Data Availability

The data are unavailable due to ethical restrictions.

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
