# Peer review of "The Role of Diastolic Stress Echo and Myocardial Work in Early Detection of Cardiac Dysfunction in Women with Breast Cancer Undergoing Chemotherapy"

_biomedicines, 2025, doi:10.3390/biomedicines13102341_

Round 1
Reviewer 1 Report
Comments and Suggestions for Authors
I reviewed with interest the manuscript by Stefanos Sokratous et al. "The role of diastolic stress echo and myocardial work in early detection of cardiac dysfunction in women with breast cancer undergoing chemotherapy". In this article, the authors analyzed the capabilities of relatively new echocardiography parameters in the diagnosis of cardiotoxicity during chemotherapy for breast cancer. The authors' data are interesting, which show that conducting Diastolic Stress Test Echocardiography allows identifying patients with diastolic dysfunction as a manifestation of therapy cardiotoxicity. This will allow identifying the development of cardiotoxicity at the earliest possible stages.
While reviewing the manuscript, I had the following comments and questions:
- I did not find references in the text of the manuscript to ref. 11-12 in the list of references.
- It is necessary to formulate the purpose of the study more clearly. The phrase "Moreover, we introduced DSTE as a screening tool for prompt diagnosis of HFpEF in this cohort" (lines 126-127) obviously cannot be the purpose of the study.
- Section 2.2. Echocardiography should provide information on the assessed parameters both at rest and during the stress test.
- It is also necessary to provide data on the method for calculating the HFA-PEFF score, since this indicator plays an important role in identifying suspected early stages of cardiotoxicity during chemotherapy.
- Not all abbreviations are deciphered in the notes to Table 2b.
- On the contrary, the notes to Table 3 provide decipherment of abbreviations that are not in the table.
- Based on the HFA-PEFF score calculations, the authors identified a group of patients with highly suspected or diagnosed HFpEF (HFA-PEFF score≥4). The data of this group were then compared with the group with CTRCD (Table 3). However, for the purposes of early diagnostics of cardiotoxicity, it is also necessary to present a comparison of this group with the group without CTRCD.
- In section 3.3. Correlations, the authors provide text results of logistic regression analysis without providing figures for the identified dependencies. This approach to presenting the results is insufficient; it is necessary to provide digital data in a generally accepted format in the form of a table (the SPSS package allows this).
- In the Discussion section, the authors consider in detail possible methods of treating cardiotoxicity during chemotherapy (lines 468-488). This section is redundant, since the article did not study various treatment methods. This text should be removed from the article
Reviewer 2 Report
Comments and Suggestions for Authors
This study focuses on the early detection of chemotherapy-related cardiac dysfunction in women with breast cancer by combining advanced echocardiographic techniques with cardiac biomarkers, using a prospective case–control design to validate their diagnostic value. The findings demonstrate that these novel approaches outperform LVEF alone in identifying subclinical cardiac injury and the potential risk of HFpEF. Overall, the study is clinically relevant and provides meaningful insights into cardio-oncology practice.
(1) With regard to methodology, although the study is prospective in design, the inclusion criteria for patients are relatively broad and lack detailed stratification according to chemotherapy regimens and cumulative doses, both of which are strongly associated with the risk of cardiotoxicity. The absence of subgroup analyses based on treatment type and dosage may limit the interpretability and generalizability of the results. Incorporating such stratification would provide a more comprehensive understanding of how specific therapeutic exposures influence cardiac outcomes.
(2) Concerning the diagnostic techniques, while GLS and DSTE demonstrated good diagnostic performance, the manuscript provides limited discussion of their reproducibility and operator dependence in real-world settings. Although the use of a single operator reduces intra-study variability, it does not fully reflect the multi-center, multi-operator conditions under which these tests would be applied in clinical practice. A more explicit consideration of feasibility and inter-observer reliability would enhance the understanding of their applicability in routine follow-up.
(3) In terms of interpretation, the authors highlight the independent predictive value of GLS, NT-proBNP, troponin, and DSTE-derived TRVmax, but the discussion does not sufficiently address how these parameters could be integrated into longitudinal clinical decision-making. Specifically, the potential thresholds for initiating cardioprotective therapy or modifying oncologic treatment remain unclear. A more detailed exploration of how these findings can be translated into practical management strategies would significantly strengthen the clinical impact of the study.
Reviewer 3 Report
Comments and Suggestions for Authors
I have appropriately reviewed the paper entitled “The role of diastolic stress echo and myocardial work in early detection of cardiac dysfunction in women with breast cancer undergoing chemotherapy”. This is a well-conceived and clinically relevant prospective case–control study addressing an important topic in cardio-oncology: early identification of chemotherapy-related cardiac dysfunction (CTRCD) and HFpEF using advanced echocardiographic techniques (GLS, myocardial work indices, DSTE) combined with biomarkers. The study design is appropriate, the sample size is acceptable, and the findings are novel with potential clinical implications. However, some issues related to clarity, methodology, statistical robustness, and interpretation of results should be revised. The introduction section should be shortened, with a greater focus on the study hypothesis. The final CTRCD group included only 28 patients, limiting the power for regression and ROC analysis. Authors should clarify whether the sample size calculation was performed a priori. The heterogeneity of chemotherapy regimens (anthracycline ± trastuzumab ± other agents) is not fully stratified in terms of outcomes.
Furthermore, there was no data available on the average dose of chemotherapy between the groups. Different agents have different cardiotoxicity risks. The diagnostic threshold used for CTRCD should be explicitly stated (absolute LVEF drop, GLS cut-off, biomarker rise?). Currently, it appears to be based on guideline criteria, but more details are needed. Multivariate models with small numbers (28 CTRCD cases) risk overfitting. Authors should confirm the number of predictors vs events and consider a more parsimonious model. The AUC for TRVmax is promising, but cross-validation or bootstrapping would strengthen reliability. Twelve months may be too short to capture late-onset CTRCD or HFpEF. Please include a subgroup analysis for predictors of CTRD in BC pts without any baseline disease, including diabetes and hypertension.
The limitations section mentions this, but should emphasize that conclusions are limited to early cardiotoxicity.The discussion effectively highlights the potential clinical use, but recommendations for integrating it into routine practice are premature. Authors should temper conclusions or clearly state that validation in multicenter cohorts is required. Abstract: The Results section should specify the size of the HFpEF subgroup (16 patients) for clarity. Table 1: TRVmax SD looks pretty high in the control group (2.58 ± 2.50 m/s) – please verify data accuracy. Some abbreviations (e.g., GCWI vs GCW) are inconsistently used. Standardization would improve readability. Additional figures of longitudinal changes in GLS, MW indices, or biomarkers would enhance clarity.
Round 2
Reviewer 1 Report
Comments and Suggestions for Authors
The authors responded to my comments and made corrections to the text of the manuscript. I have no other comments.
